# STAT3-mediated allelic imbalance of novel genetic variant Rs1047643 and B-cell-specific super-enhancer in association with systemic lupus erythematosus

**Yanfeng Zhang[1]\*[†], Kenneth Day[2], Devin M Absher[1]\***

[1]HudsonAlpha Institute for Biotechnology, Huntsville, United States; [2]Zymo Research Corp, Irvine, United States

**\*For correspondence:**
yanfengzhang1984@outlook.com (YZ);
dabsher@hudsonalpha.org (DMA)

**Present address:** [†]Department of Population and Data Sciences, University of Texas Southwestern Medical Center, Dallas, United States

**Competing interest:** The authors declare that no competing interests exist.

**Abstract** Mapping of allelic imbalance (AI) at heterozygous loci has the potential to establish links between genetic risk for disease and biological function. Leveraging multi-omics data for AI analysis and functional annotation, we discovered a novel functional risk variant rs1047643 at 8p23 in association with systemic lupus erythematosus (SLE). This variant displays dynamic AI of chromatin accessibility and allelic expression on *FDFT1* gene in B cells with SLE. We further found a B-cell restricted super-enhancer (SE) that physically contacts with this SNP-residing locus, an interaction that also appears specifically in B cells. Quantitative analysis of chromatin accessibility and DNA methylation profiles further demonstrated that the SE exhibits aberrant activity in B cell development with SLE. Functional studies identified that STAT3, a master factor associated with autoimmune diseases, directly regulates both the AI of risk variant and the activity of SE in cultured B cells. Our study reveals that STAT3-mediated SE activity and cis-regulatory effects of SNP rs1047643 at 8p23 locus are associated with B cell deregulation in SLE.

## Editor's evaluation

Through extensive computational data analyses and functional experiments, Zhang and colleagues reported a novel functional germline variant at the risk locus 8p23 for systemic lupus erythematosus. They provided evidence that the observed risk association in this locus may be mediated through the variant regulating a nearby susceptibility gene. This study advances the understanding of the genetic susceptibility and biology underlying systemic lupus erythematosus.

## Introduction

Super-enhancers (SEs) are recently discovered large domains of clustered enhancers (*Parker et al., 2013*; *Whyte et al., 2013*). The extraordinary feature of SEs is the extremely high and broad enrichment of enhancer-related transcription factors (TFs), H3K4me1 and H3K27ac modifications, resulting in high capability to enhance gene expression programs (*Whyte et al., 2013*). A large quantity of SEs show cell/tissue specificity (*Vahedi et al., 2015*), thereby they have become principal determinants of cell identity (*Hnisz et al., 2013*). Nonetheless, disease-associated SEs, in particular those exhibiting aberrant activity in autoimmune diseases, are less characterized.

Signal transducer and activator of transcription 3 (STAT3), as one of seven STAT family members, is activated by phosphorylation at tyrosine 705 (Y705) and/or at serine 727 (S727) (*Decker and Kovarik, 2000*). After import to the nucleus, the phospho-STAT3 (pSTAT3) modulates gene transcription by binding its target sequence (*Levy and Darnell, 2002*). STAT3 has gained broad attention because it

plays a key role in a variety of pathophysiological immune responses related to lymphocyte development and differentiation, and in other cellular processes of normal and tumor cells (*Yu et al., 2009*).

Systemic lupus erythematosus (SLE) is an autoimmune disease that is known to be associated with an array of abnormal immune cell signaling. B-cell hyperactivity in auto-antigen recognition and interaction with T-cells, which ultimately results in multi-organ damage, is central to the pathogenesis of SLE (*Rahman and Isenberg, 2008*). Genetic factors conferring a predisposition to the development of SLE have been widely characterized. Over 100 loci have been implicated in SLE by genome-wide association studies (GWAS) (*Catalina et al., 2020*; *Yin et al., 2021*). Among them, several genes and/or loci are potent as putative drivers of the disease. For example, genetic risk variants at the promoter of *BLK* at 8p23 locus alter *BLK* transcription activity and thus contribute to autoreactive B-cell responses (*Guthridge et al., 2014*). Nonetheless, the GWAS-identified genetic variants together explained approximately 30% of the heritability of SLE (*Sun et al., 2016*; *Morris et al., 2016*), suggesting a requirement of further efforts to explain the missing heritability of SLE. Meanwhile, there is growing evidence that genetic risk factors behave in a context-dependent or cell-specific manner (*Guthridge et al., 2014*; *Gallagher and Chen-Plotkin, 2018*). Thus, for SLE and other autoimmune diseases, there is a need to identify the regulatory programs in which these genetic factors impact the immune cell developmental processes.

One approach for tying genetic risk to function in the post-GWAS era (*Gallagher and Chen-Plotkin, 2018*), is a measurement of allelic imbalance (AI) on two alleles at a given heterozygous locus, typically at single-nucleotide polymorphism (SNP). The genes and/or loci with SNPs exhibiting AI could provide a strong foundation for implicating the genetic or epigenetic mechanisms linked to complex traits or diseases (*Pastinen and Hudson, 2004*; *Yan et al., 2002*). As a readout of AI, analyses of allele-specific chromatin accessibility and allele-specific RNA expression have accumulated a wealth of interesting findings, including functional cis-regulation (*Li et al., 2013*; *Zhang et al., 2020b*), genomic imprinting (*Pollard et al., 2008*), X-chromosome inactivation or escape (*Zhang et al., 2020a*). Therefore, tracking AI difference in a comparison between diseases and controls may enable to uncover novel functional variants associated with complex diseases. In this study, we describe one such strategy through integrative multi-omics analysis to discover known or novel functional variants associated with SLE, and report on the identification of a novel risk variant rs1047643 and B-cell-specific SE in B cells with SLE. We further demonstrate that the resultant allelic imbalanced variant and SE activity are directly controlled by STAT3, a master TF that plays a critical role in B cell development and highly associates with autoimmune diseases.

## Results

### Multi-omics data summary

Functional genomics sequencing data sets comprising 279 samples from eleven studies were collected (*Supplementary file 1*). Of eleven studies, seven are SLE case-control studies with data across three immune cell types including B cells, T cells, and Neutrophils (*Supplementary file 2*). Also included in the present study were SNP microarray data from a SLE GWAS study (n = 2279).

### Identification of SLE-associated variant showing AI at both chromatin and RNA levels

We next designed a two-stage study (*Figure 1*) to identify putative SLE-associated functional variants. In stage I, also named as the discovery stage, two chromatin accessibility (ATAC-seq) data sets (Accession ID: GSE118253 and GSE71338, *Supplementary file 1*) comprised 49 samples were analyzed. We focused on those variants displaying difference in AI of chromatin accessibility at heterozygous SNP sites in a comparison between SLE and controls (see Materials and methods in detail). In total, 18,456 (the GSE118253 dataset) and 4319 (the GSE71338 dataset) SNPs were tested. We collected the resulting SNPs with nominal (unadjusted) p < 0.05. From the reciprocal validation between two data sets, three SNPs (rs1047643, rs246367, and rs72642993) were identified to show the significant AI (combined p < 0.01) in B cells from patients with SLE, relative to controls (*Figure 2A* and *Figure 2—figure supplement 1*). Based on annotations from the GWAS central (*Beck et al., 2020*) and HaploReg (*Ward and Kellis, 2016*) databases, we then focused on the rs1047643 because it is the most promising variant. Meanwhile, in B cells at different stages, the allelic preference of chromatin accessibility for SNP rs1047643 is

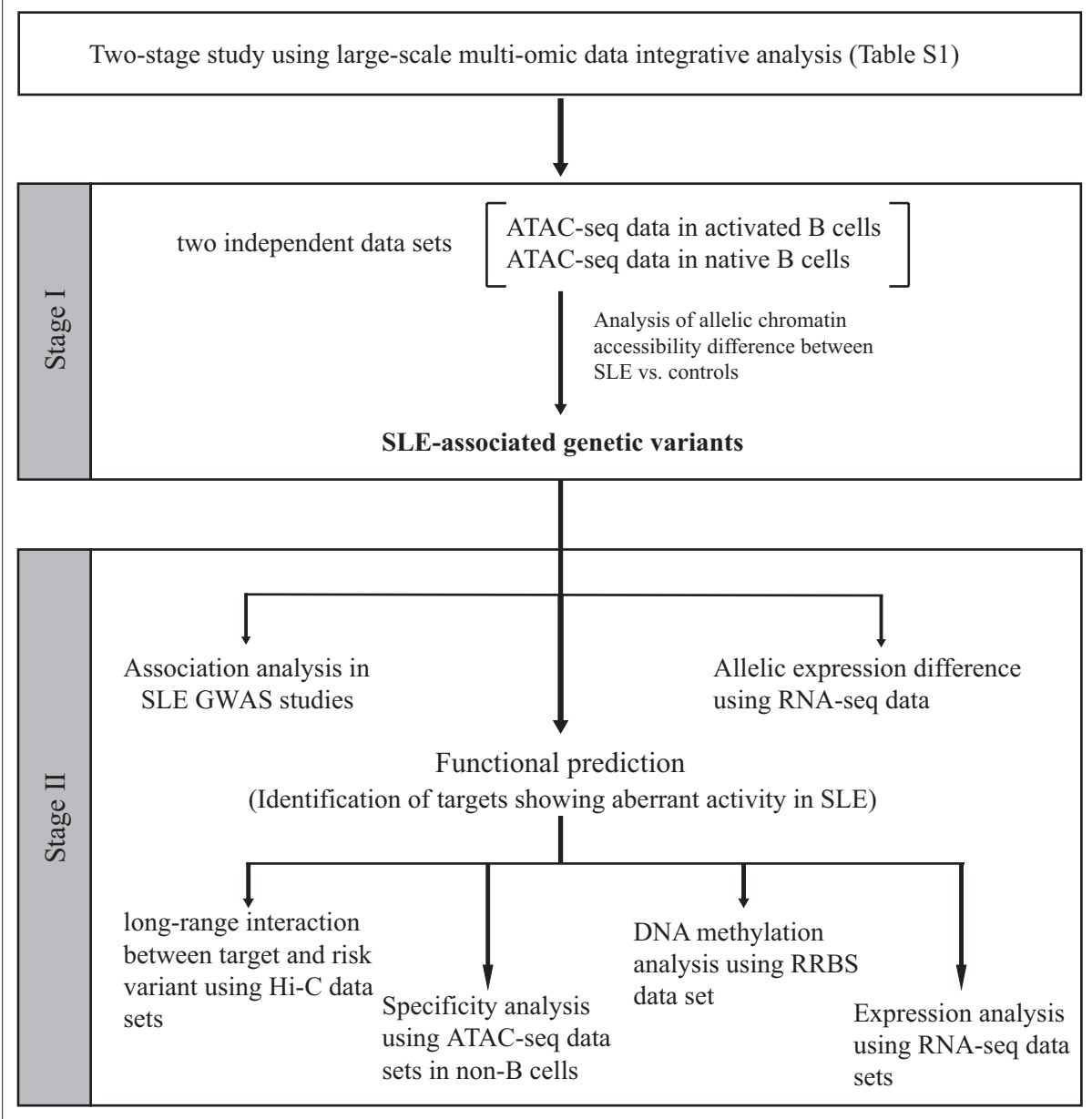

**Figure 1.** Schematic of the study design. On the basis of the functional genomic data feature, a two-stage study was designed. Summary of data sets is available in ***Supplementary files 1-2***.

alterable. For example, the T allele exhibits more preferential chromatin accessibility in activated B cells from patients, relative to the C allele. However, the direction is reversed in SLE naive B cells.

Because the rs1047643 is located in the first exon of *FDFT1* gene (***Figure 3E***), it enables us to test the functionality of this variant at the transcriptional level. Analyzing RNA-seq data (Accession ID: GSE118254), we determined the AI of RNA transcripts for the rs1047643 by comparing the AR values (see Materials and methods in detail) between SLE and controls. In line with results shown above, we observed the dynamic AI pattern on the transcriptional level for the rs1047643 (***Figure 2B***). Meanwhile, this dynamic allelic expression pattern is specific during B cell development with SLE (***Figure 2C***).

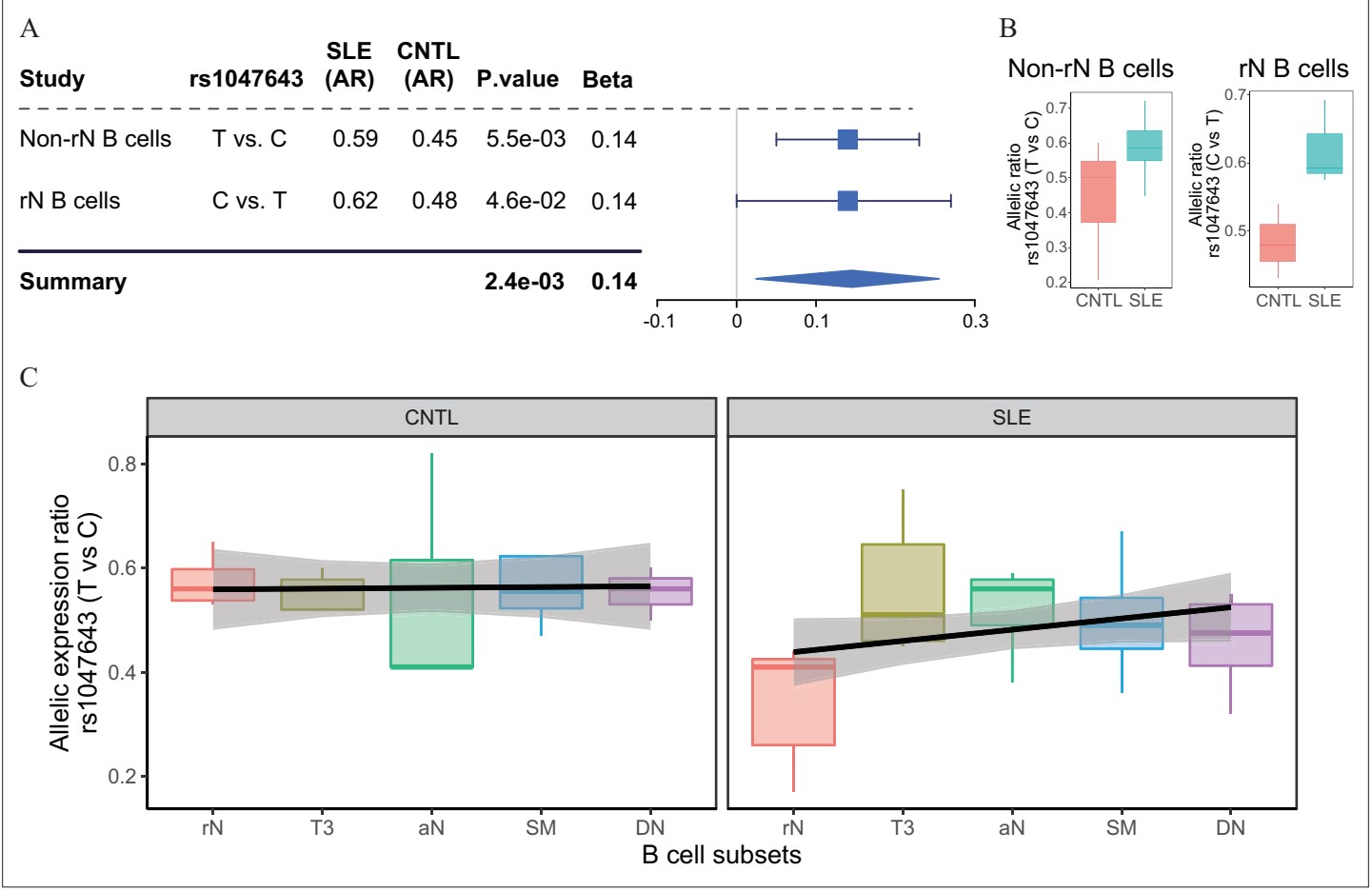

**Figure 2.** Change of allelic chromatin accessibility and expression in B cell subtypes from SLE patients and controls. (**A**) Forest plot showing AI of allelic chromatin state of SNP rs1047643 in both resting naive (rN) and activated (Non-rN) B cells in patients of SLE compared with healthy controls. The p-value per study and combined p-value (summary) are calculated based on the linear regression model and Fisher's method, respectively. The plot in the right panel displays the 95% of confidence interval of beta-value. (**B–C**) Boxplots showing allelic expression of SNP rs1047643 in both rN and activated B cells in patients with SLE as compared with healthy individuals. All raw data are available in *Figure 2—source data 1*.

The online version of this article includes the following source data and figure supplement(s) for figure 2:

**Source data 1.** Source files for presenting results in *Figure 2*.

**Figure supplement 1.** Change of allelic chromatin accessibility in B cell subtypes from SLE patients and controls.

**Figure supplement 2.** Expression pattern of FDFT1 and BLK across B cell subtypes in patients with SLE and healthy controls.

**Figure supplement 3.** Expression pattern of FDFT1 and BLK across B cell subtypes in patients with SLE and healthy controls.

## Association with SLE risk in American Hispanic and European populations

Because SNP rs1047643 has not been reported to be associated with the susceptibility of SLE and other autoimmune diseases, we next tested the association using a dataset from an SLE GWAS case-control study. Employing the univariate analysis for SNP rs1047643 in samples from Hispanic populations, we identified an association of the rs1047643 with SLE risk (OR per C effect allele = 0.77, 95% CI 0.54–0.99, p = 0.023 after adjusting for covariates, *Figure 3A*), albeit not reaching the significance after adjustment for 12 GWAS index SNPs (the top track in *Figure 3E*, where one SNP rs2736336 is excluded due to its multivariate alleles). We also examined an association in European population (n = 19,468) from *Bentham et al., 2015* study. A similar result was observed for the rs1047643 (OR per C effect allele = 0.91, 95% CI 0.84–0.98, p = 0.02, *Supplementary file 3*).

An analysis of linkage disequilibrium (LD) with each of 12 GWAS tag SNPs showed that there was no strong LD ($r^2$ <0.1) between SNP rs1047643 and GWAS SNPs in European population (*Supplementary*

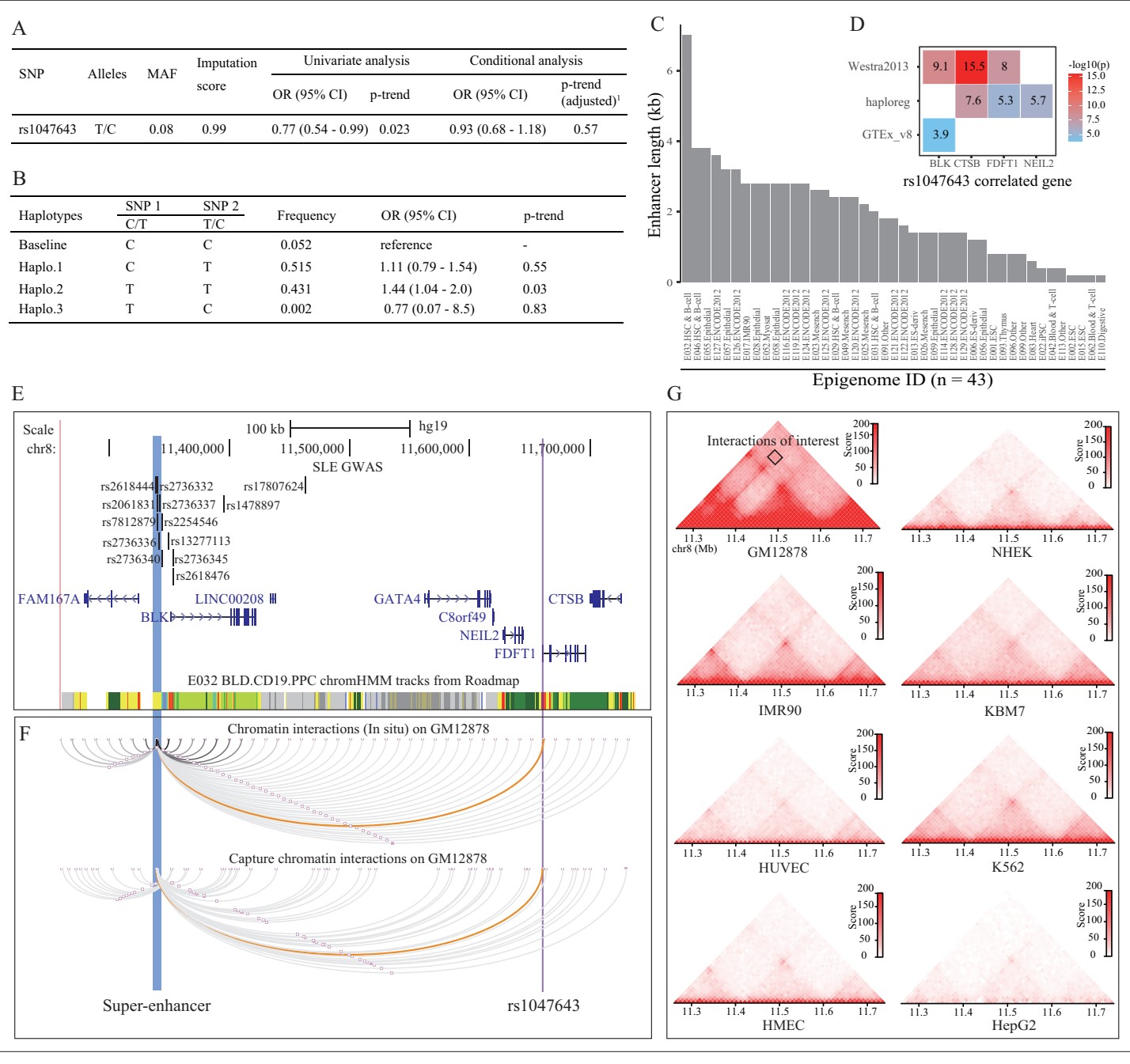

**Figure 3.** Association analysis and functional prediction of SNP rs1047643. (**A**) Association results for the SNP rs1047643 with SLE risk in single marker analyses. MAF, minor allele frequency; OR, odds ratio; CI, confidence interval. Adjusted p-trend: after adjustment for 12 GWAS index SNPs (shown in **E**) in a logistic regression model. (**B**) Haplotype analyses of the two SNPs (SNP1: GWAS indexed SNP rs17807624; SNP2: rs1047643) in relation to SLE risk. Baseline (the reference haplotype) represents the alleles associated with a reduced risk in two SNPs. (**C**) Barplot showing the genomic length of chromHMM-annotated enhancer state on the super-enhancer region (blue highlighted in 3 C) in 43 epigenomes. (**D**) Plot shows the eQTL result of SNP rs1047643 in whole blood or B cells from three databases (shown in y-axis). (**E**) Genomic annotations of the SNP rs1047643. The three tracks show locations of 13 GWAS index SNP, gene annotation and 15-state chromatin segments in CD19+ B cells at 8p23 locus, respectively. Vertical blue and purple lines, represents the location of super-enhancer and SNP rs1047643, respectively. (**F**) Long-range interaction between a super-enhancer and SNP rs1047643. The two tracks show chromatin interactions from two independent studies using whole-genome Hi-C and capture Hi-C technologies, respectively. Orange curves show the interactions between the super-enhancer and the SNP rs1047643. (**G**) Heatmaps showing the 3D DNA interactions at 8p23.1 locus in eight cell lines. The rectangle represents interactions between the super-enhancer and the SNP rs1047643. All raw data are available in *Figure 3—source data 1*.

The online version of this article includes the following source data and figure supplement(s) for figure 3:

*Figure 3 continued on next page*

*Figure 3 continued*

**Source data 1.** Source files for presenting results in *Figure 3*.

**Figure supplement 1.** Chromatin interactions with *FDFT1* promoter region (marked in green arrow) on 8p23 locus from CHi-C data with duplicates in two types of normal T cells.

**Figure supplement 2.** Heatmaps of Long-range chromatin interactions from Hi-C data in 8p23 locus at 10 kb (or 20 kb) resolution in a panel of human tissues (n = 9) from the 3D Genome Browser.

*file 4*). This result indicates that SNP rs1047643 is a potentially SLE GWAS independent functional variant. Of the 12 index SNPs, indeed, one index SNP rs17807624 with the statistical significance with $p < 1.5 \times 10^{-3}$ using the univariate analysis, is the top signal to which the SNP rs1047643 is conditional. Thus, we performed haplotype analyses on these two SNPs (index SNP rs17807624 and rs1047643, *Figure 3B*). Compared with the reference haplotype, which carries the alleles associated with a reduced risk in two SNPs, haplotype 2, which carries the risk-associated alleles, showed a significant association (p = 0.03).

## Functional annotation

An analysis of eQTL data derived from three independent cohorts indicated both proximal ( < 200 kb) and distal ( > 200 kb) regulatory potential for the SNP rs1047643 in normal B or blood cells (*Figure 3D*). Interestingly, besides correlated with three adjacent genes (*FDFT1*, *CTSB* and *NEIL2*), the rs1047643 is also an eQTL linked with an upstream *BLK* gene in a distance of ~300 kb, a result that is detected in two independent data sets. An analysis of RNA-seq data from two independent studies (Accession ID: GSE118254 and GSE92387, *Supplementary file 1*) consistently showed that expression patterns for two representative genes (*BLK* and *FDFT1*) are gradually increased in a developmental process from naive to memory B cells, in particular, the double negative memory B cell subset in patients with SLE, the pattern that is not observed in controls (*Figure 2—figure supplements 2 and 3*).

By searching for enhancers and other regulatory elements across 8p23 locus from a dataset of the 127 epigenomes from Roadmap, we identified a SE with a length of 7 kb in the upstream of *BLK* gene in CD19+ B cells (Epigenome ID: E032, *Figure 3E*). An analysis of enhancer elements across the 127 epigenomes showed 43 (33.9%) epigenomes had enhancers at this SE region. Comparative analysis of the enhancer length at this SE region on the 43 epigenomes further showed that this SE is specific in CD19+ B cells (Epigenome ID: E032, *Figure 3C*).

Analyzing Hi-C data sets from two independent studies in GM12878 cells, we observed a DNA looping between the SNP rs1047643 within FDFT1 and the SE region (*Figure 3F*). More importantly, in GM12878 B-lymphoblastic cells, this SE region has a wealth of long-range interactions with adjacent genes (e.g., BLK) and functional elements. In contrast, in another seven cells (*Figure 3G*), as well as in normal T cells (*Figure 3—figure supplement 1*) and nine selected tissues (*Figure 3—figure supplement 2*), these interactions are either much weaker or completely absent. These results indicate that the physical interaction between SNP rs1047643 and SE region, and many interactions with this SE, are specific to B-lymphocytes.

## Specificity in B cells

We then hypothesized that the SE region may show aberrant activity in B cells from SLE patients. To test this hypothesis, we conducted quantitative analysis on the same ATAT-seq data (Accession ID: GSE118253 and GSE71338) used in stage I (see Materials and methods in detail). Comparison of SE activity in a quantitative manner between SLE patients and controls indicated that the SE activity is gradually increased through B cell development in SLE patients (*Figure 4A–B*), with a hyper-activity being observed in double negative (DN) B cells in patients, relative to controls (*Figure 4B–C*). It should be noted that such a pattern is not observed in the background sampling (*Figure 4—figure supplement 1*). Similarly, the rs1047643-containing promoter activity also shows up-regulation toward B cell development in SLE patients (*Figure 4D–E*). In a comparison of B cell development on activities of SE and FDFT1 promoter regions in two individuals, the chromatin accessibility on both regions in an individual with SLE is increased during B cell development, but remains relatively unchanged in the healthy individual (*Figure 4F–H*).

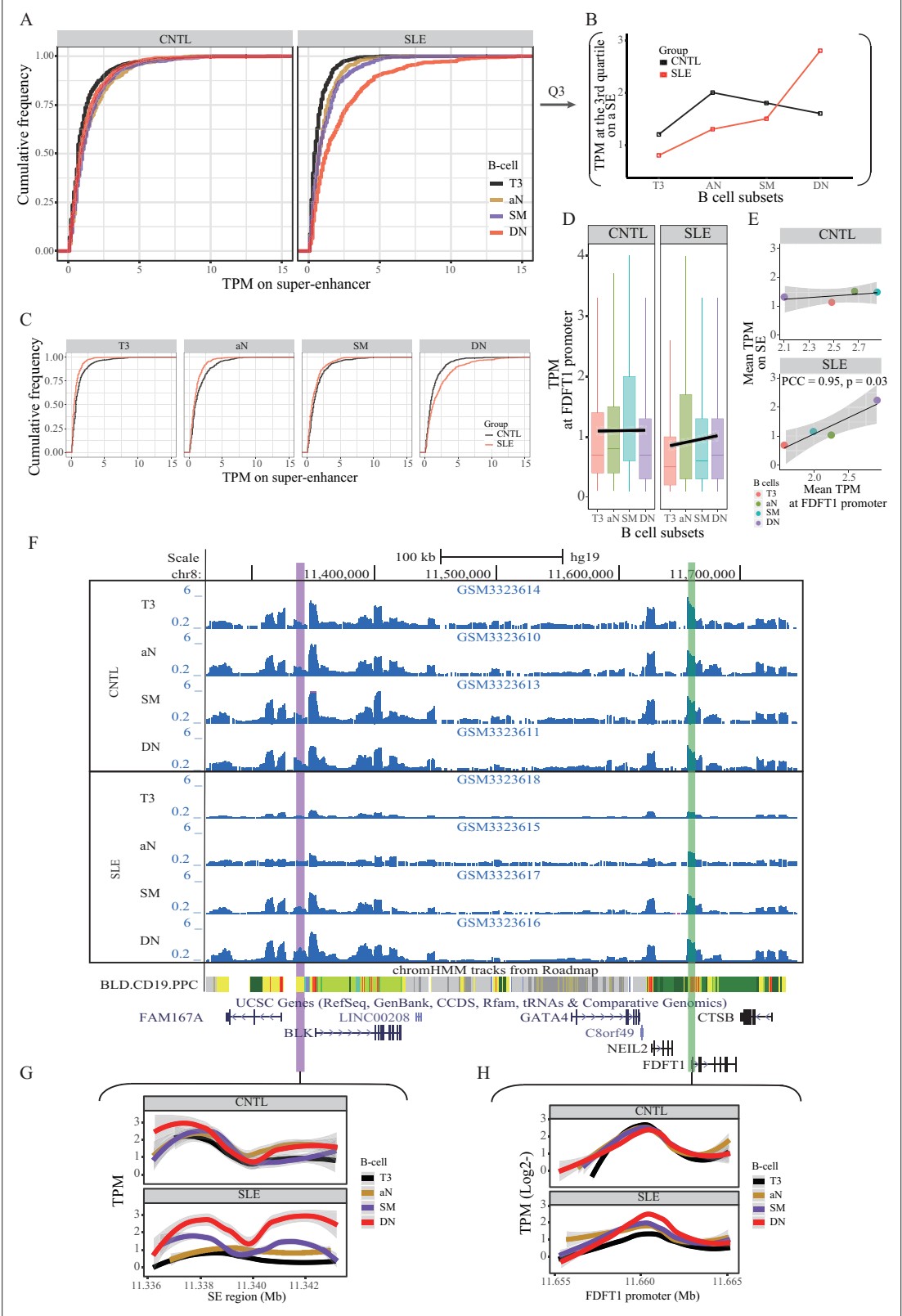

**Figure 4.** Aberration of super-enhancer and *FDFT1* promoter region in B cell subtypes from SLE patients. (**A**) Empirical cumulative distribution of TPM values per 50 bp window across the 7 kb SE region in B cell subsets for disease and control groups. (**B**) Plots showing the TPM values at the third quartile (**Q3**) across B cell subtypes as a comparison between SLE and controls. (**C**) Empirical cumulative distribution of TPM values on the SE region (same as shown in A) in a comparison between two groups across four B cell subtypes. (**D**) Boxplots showing the TPM values per 50 bp window at the

*Figure 4 continued on next page*

*Figure 4 continued*

*FDFT1* promoter region in B cell subtypes for SLE and controls. The black lines and grey areas represent the linear regression results towards the B cell development from T3 to DN stages, and 95% of CI. (**E**) Plots showing the correlation between super-enhancer and *FDFT1* promoter regions based on mean TPM values with respect to B cell subtypes in SLE and controls. (**F**) Wiggle plot showing the enrichment of open chromatin states at 8p23.1 locus in B cell subtypes for two individuals (a healthy individual at upper panel, and a patient with SLE at lower panel). Purple and green vertical lines represent the locations for super-enhancer and *FDFT1* promoter, respectively. Quantitative comparison of chromatin accessibility states in SE (**G**) and *FDFT1* promoter regions (**H**) with respect to B cell subtypes. All raw data are available in ***Figure 4—source data 1***.

The online version of this article includes the following source data and figure supplement(s) for figure 4:

**Source data 1.** This txt file contains source data used for the quantitative analyses shown in ***Figure 4***.

**Figure supplement 1.** Genome-wide background analysis of ATAC-seq data.

**Figure supplement 2.** Aberration of super-enhancer in resting naive B cell subtypes from SLE patients in relation to healthy controls.

**Figure supplement 3.** Super-enhancer activity in T and neutrophils from SLE patients and controls.

We also quantitatively compared open chromatin states of SE and FDFT1 promoter regions in resting naive B cells (Accession ID: GSE71338). Concordant with the results from active B cell subsets, the open chromatin states on both regions are low in non-active B cells from SLE patients, relative to healthy controls (***Figure 4—figure supplement 2***).

We further conducted quantitative analyses on ATAC-seq data from another two independent studies in two immune cell types, T cells, and neutrophils (Accession ID: GSE139359 and GSE110017, ***Supplementary file 1***). The results showed that there was no marked enrichment of ATAC-seq reads on both the SE and FDFT1 promoter regions in these two immune cell types for both SLE and controls (***Figure 4—figure supplement 3***). Collectively, these results suggest a B cell specific, rs1047643-interacting SE whose activity is aberrant in SLE B cell development.

## Hypomethylation in SLE B cells

We further analyzed DNA methylation in the SE region using RRBS data in B cell development in a comparison between SLE and controls (Accession ID: GSE118255, ***Supplementary file 1***). Our results show that DNA methylation levels on the SE region are gradually decreased in the developmental process from resting native (rN) to memory B cells in patients with SLE (***Figure 5A***). In contrast, there is no such obvious change of DNA methylation pattern in the control group. In addition, an analysis of DNA methylation levels on the background sampling doesn't show such a difference between SLE and controls (***Figure 5—figure supplement 1***), further suggesting that epigenetic change on the SE region is biologically meaningful in B cell development in SLE.

A correlation analysis also showed a marked negative correlation between open chromatin states (TPM values, also presented in ***Figure 4E***) and DNA methylation levels at the SE region in the SLE group, relative to the healthy controls (***Figure 5B***). Together, these results reinforce the aberrant activity of SE in developmental process of B-lymphocytes in patients with SLE.

## STAT3 binding on both super-enhancer and rs1047643-residing regions

TF-motif enrichment and binding analysis using the ENCODE TF ChIP-seq dataset (v3) predicted that STAT3 may bind to both the SNP rs1047643-containing promoter and SE regions (data not shown). To validate the finding, we designed two pairs of primers (SE5 and SE3, ***Figure 6A***) to determine the STAT3 binding on SE region and its contribution to the SE activity using STAT3, H3K4me1 and H3K27ac ChIP-qPCR assays in GM11997 cells. Under normal culture conditions, we validated that pSTAT3, H3K4me1 and H3K27ac modifications are remarkably enriched on the SE region (***Figure 6A, B and D***), but not on the negative control region (***Figure 6—figure supplement 1***) in B-lymphoblastic cells, relative to IgG mock controls. We then conducted both the inhibition and activation of STAT3 DNA-binding activity using two small molecules. In B-lymphoblastic cells challenged with S3I-201, a STAT3 DNA binding inhibitor, both the DNA binding of STAT3 on SE region and the SE activity are significantly reduced (***Figure 6A and B***), relative to control. In GM11997 cells treated with ML115, a selective activator of STAT3 (41), both the STAT3 DNA-binding capability on SE region and the SE activity are significantly increased (***Figure 6D and E***), relative to controls. These results together demonstrate that STAT3 directly modulates the SE activity.

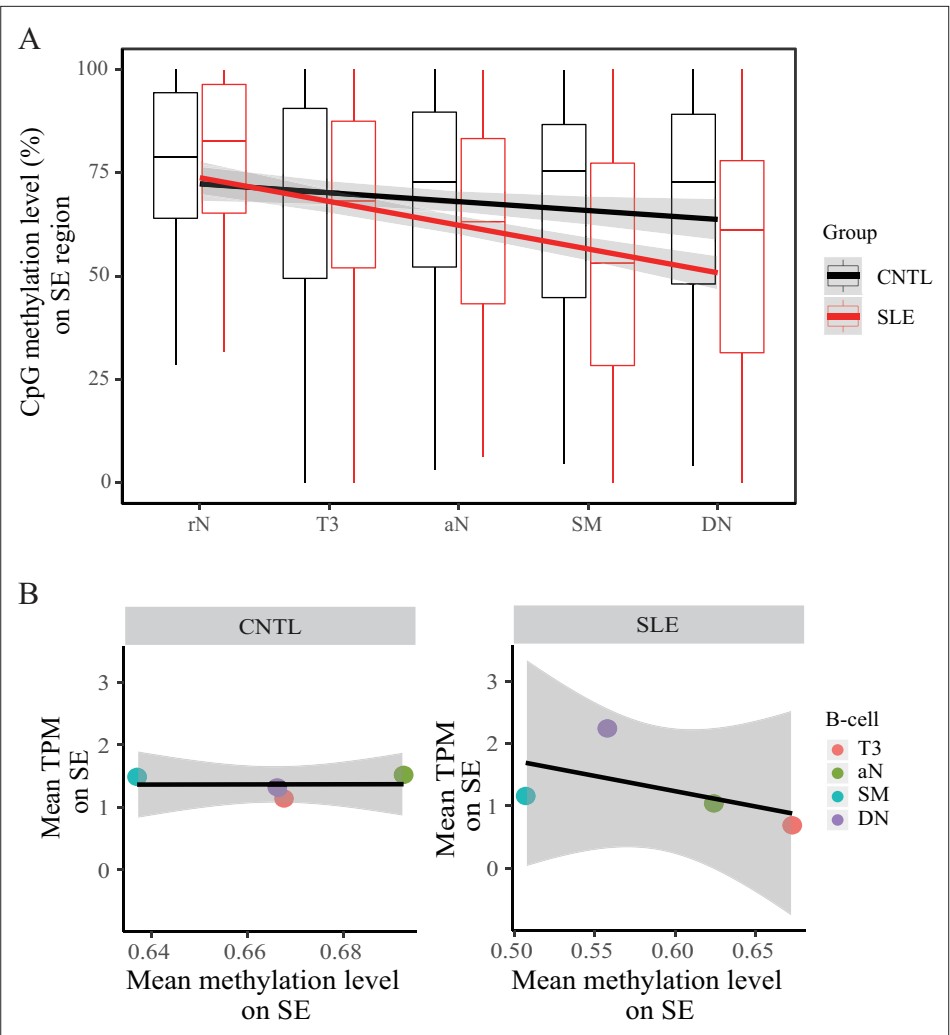

**Figure 5.** Hypomethylation in super-enhancer region in B cell subtypes from SLE patients. (**A**) Boxplots showing the CpG methylation levels per 50 bp window in 7 kb SE region in B cell subtypes for SLE and control groups. The black and red lines represent the linear regression results towards the B cell development from rN to DN stages for SLE and controls, respectively. (**B**) Plots showing the correlation between TPM values (y-axis) and DNA methylation levels (x-axis) averaged over each B cell type in SLE and controls. All raw data are available in ***Figure 5—source data 1***.

The online version of this article includes the following source data and figure supplement(s) for figure 5:

**Source data 1.** This txt file contains source data used for the quantitative analyses shown in ***Figure 5***.

**Figure supplement 1.** DNA methylation comparison across randomly selected regions in B cell subtypes between patients with SLE and controls.

---

We next tested whether the STAT3 might also regulate the rs1047643-residing regions. Using allelic qPCR assay, we confirmed that genomic DNA in the GM11997 cells carries a heterozygous variant for the SNP rs1047643 (***Figure 6—figure supplement 2***), enabling the AI analysis in this cell model. In GM11997 cells treated with the STAT3 inhibitor S3I-201, STAT3 binding on the risk allele T is significantly reduced, relative to the rs1049643-C allele (***Figure 6C***). Conversely, we observed an increase of STAT3 DNA binding at the rs1049643-T allele in cells stimulated with the STAT3 activator ML115 (***Figure 6F***). We further confirmed the findings in cells treated with Cucurbitacin I or IL-6 that acts an inhibitor and stimulator (***Figure 6—figure supplement 3***) of the Janus Kinase (JAK)/STAT3 signaling pathway (***Blaskovich et al., 2003***; ***Hunter and Jones, 2015***), respectively.

Consistent with findings in the STAT3 DNA-binding study, we observed a significant change of the rs1049643-T allele at the transcriptional level after treatment with S3I-201 and ML115 at different

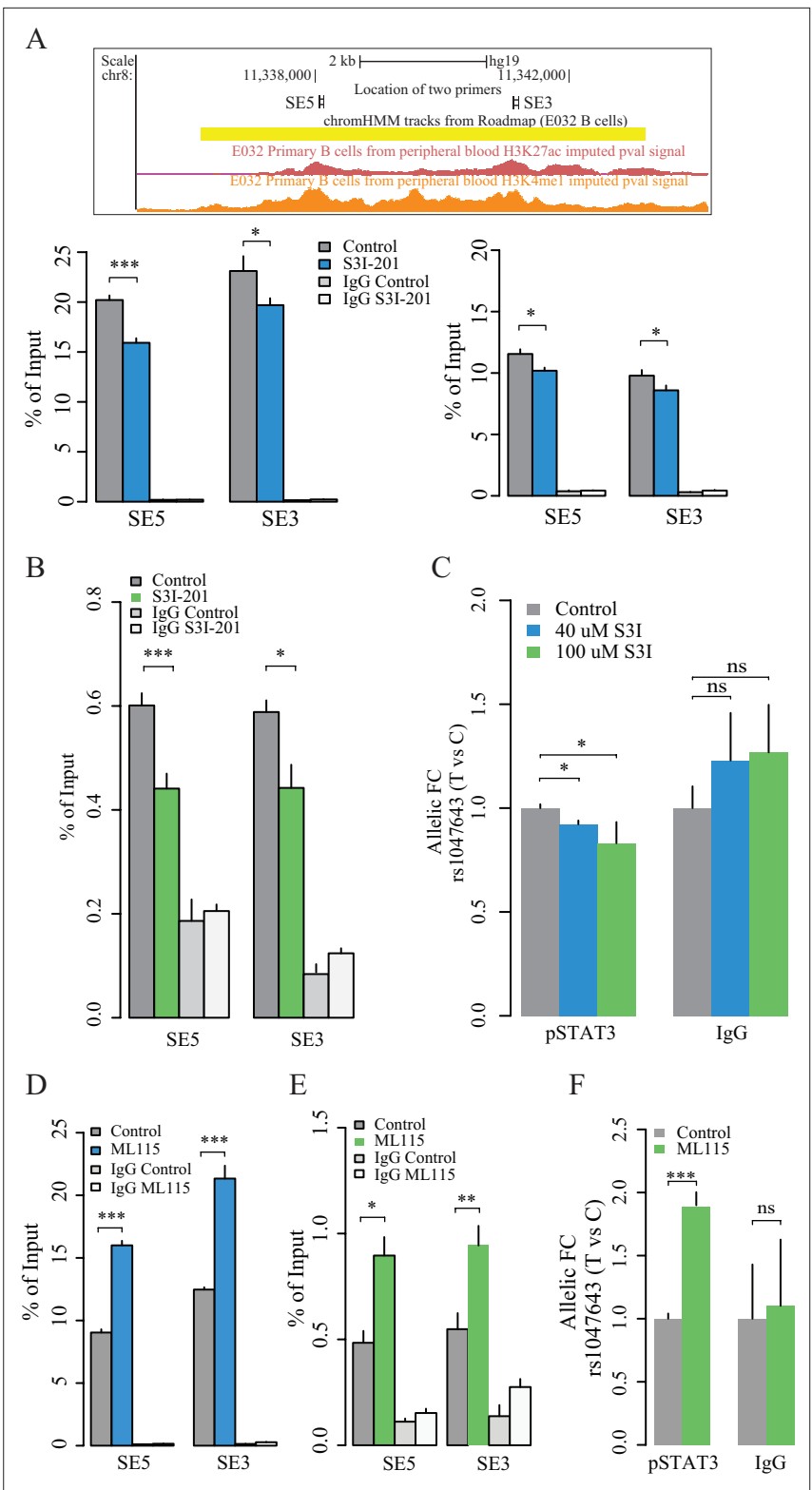

**Figure 6.** Contribution of STAT3 modulates the enhancer activity and SNP-residing locus in cultured GM11997 cells. (**A**) ChIP-qPCR for H3K27ac (left lower panel), H3K4me1 (right lower panel) and pSTAT3 (**B**) at 8p23 super-enhancer region following 40 µM S3I-201 treatment for 24 hr. Upper panel: UCSC genome browser showing the location of two pairs of qPCR primers (SE5 and SE3) on the SE region (yellow). Two tracks shown below are the enrichment of H3K27ac and H3K4me1 across the SE region. (**C**) Allelic ChIP-qPCR for pSTAT3 binding on rs1047643

*Figure 6 continued on next page*

*Figure 6 continued*

(T vs C alleles) following S3I-201 treatment for 24 hr. (**D–E**) ChIP-qPCR for H3K27ac (**D**), and pSTAT3 (**E**) at 8p23 super-enhancer region following 100 nM ML115 treatment for 6 hr. (**F**) Allelic ChIP-qPCR for pSTAT3 binding on rs1047643 in cells that have been challenged with ML115 for 6 hr as indicated. Note: the fold changes for the rs1047643-associated *BLK* and *FDFT1* genes in response to small molecules compared to vehicle (0.1% DMSO) as control, which was set as one in all cases, are presented. NS, not significance; *, p < 0.05; **, p < 0.01; ***, p < 0.005.

The online version of this article includes the following figure supplement(s) for figure 6:

**Figure supplement 1.** Quality control of ChIP experiments in GM11997 cells.

**Figure supplement 2.** Genotyping of SNP rs1047643 in GM11997 genomic DNA using allelic qPCR analysis.

**Figure supplement 3.** Validation of STAT3-mediated allelic binding in GM11997 cells.

concentrations and duration times, relative to the C allele (*Figure 7A and C* and *Figure 7—figure supplement 1*). These results suggest that the risk rs1049643-T allele is preferentially bound and regulated by STAT3 in B cells.

Finally, we determined RNA expression of BLK and FDFT1, two representative genes that correlate with the risk rs1047643. The expression levels of both genes are decreased with the treatment of

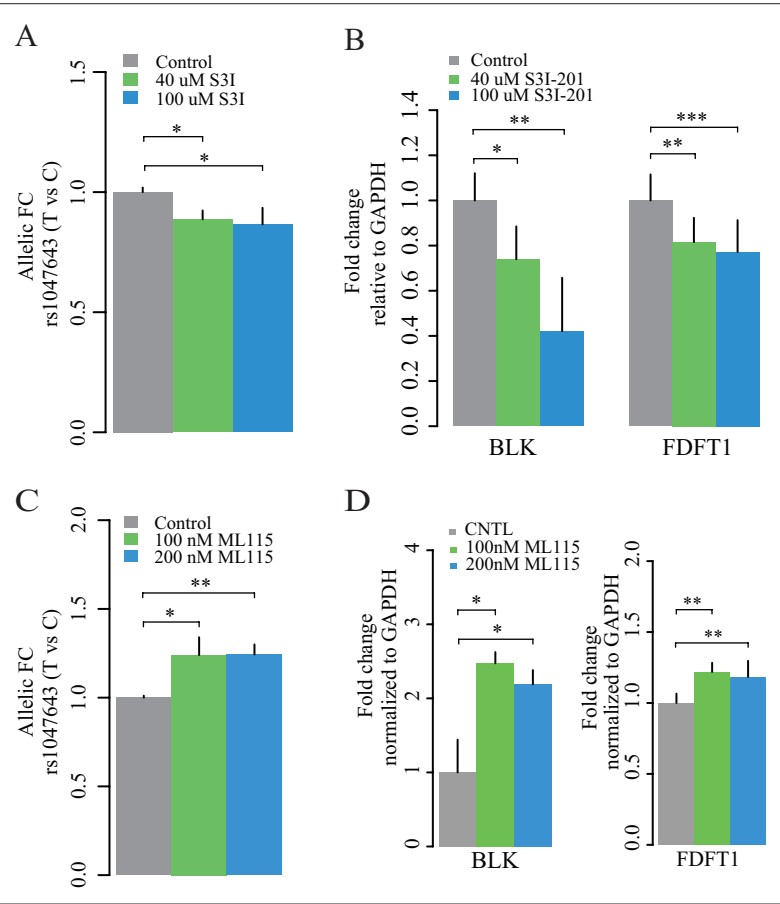

**Figure 7.** Expression of two alleles on SNP rs1047643 and its linked genes in cultured cells. Left panel: allelic RT-qPCR on SNP rs1047643 (T vs C alleles) following S3I-201 (**A**) and ML115 (**C**) treatment for 24 hr, respectively. Right panel: RT-qPCR analysis showing the fold changes for the rs1047643-associated BLK and FDFT1 genes in response to different concentrations of S3I-201 (**B**) and ML115 (**D**) compared to vehicle (0.1% DMSO) as control, which was set as one in all cases, are presented. *, p < 0.05; **, p < 0.01; ***, p < 0.005.

The online version of this article includes the following figure supplement(s) for figure 7:

**Figure supplement 1.** Expression of two alleles on SNP rs1047643 in B-lymphoblastic cells.

S3I-201 (*Figure 7B*), and upregulated with the STAT3 activator ML115 (*Figure 7D*). Together, these results suggest the STAT3-binding risk allele T is associated with increased expression of *BLK* and *FDFT1*.

## Discussion

In the present study, by integrating a variety of functional genomic data, we performed AI analysis to uncover novel functional promising variants and their regulatory targets in association with SLE. Of note, the diversity of genomic data types from this comprehensive data collection for autoimmune diseases allowed us to develop an approach not used before for accessing the role of variants in SLE disease activity.

One of the most significant findings is the identification of a novel risk variant rs1047643. The association study shows that the rs1049643-T is a risk allele for SLE. Our AI analyses indicate that the rs1049643-T allele resides in more open chromatin state and has higher expression in SLE memory B cell subsets, relative to the C allele. Functional study further provides evidence that the rs1049643-T allele is preferentially bound by STAT3. The SNP rs1047643 is also an eQTL linked with both proximal and distal genes, including *BLK*, the gene that plays a critical role in B lymphocyte development (*Saijo et al., 2003*). These results demonstrate that this novel SLE-associated risk rs1047643 whose functionality is mediated by STAT3, may play a role in allele-specific control of adjacent genes at 8p23 locus in B cells. Despite no report for association with other autoimmune diseases, this SNP has been associated with multiple myeloma (*Van Ness et al., 2008*) and follicular lymphoma (*Skibola et al., 2008*), two malignant diseases whose pathogenesis is partially associated with the dysfunction of B cells. Specifically, hyperactive STAT3 has been reported to be associated with poor survival in both diseases (*Huang et al., 2013*; *Jung et al., 2017*). Therefore, our findings may provide a clue for genetic and mechanical studies on those B cell associated diseases.

Another intriguing finding in this study is the identification of an aberrant activity of a SE in lupus B cell subsets, particularly the hyperactivity in memory B cells. In contrast, there is no enhancer activity in other immune cells (T cells and neutrophils analyzed in this study) in patients with SLE. We also demonstrate that the aberrant activity of the SE can be mediated by STAT3. Some studies have consistently reported a critical role of STAT3 in the B cell maturation, differentiation, as well as the autoimmunity (*Avery et al., 2010*; *Ding et al., 2016*). These reports further support the significance of STAT3-mediated SE aberration in B cells with SLE.

Several studies have highlighted the 8p23 locus as a major SLE susceptibility region (*Demirci et al., 2017*). Our study further expands the significance at this locus. For example, our study and others together suggest that there are a few cis-eQTLs linked with transcriptional levels of BLK (*Guthridge et al., 2014*; *Demirci et al., 2017*). Thus, we speculate that the 8p23 locus may play functional roles in B cell development in both genetic and epigenetic fashions. Besides the SNP rs1047643 discovered in the present study, there are 13 SLE-associated GWAS leading SNPs reported in this locus. Of 13 SNPs, six SNPs (*Figure 3E*) directly sit in the SE region, suggesting these risk variants may play roles in a genetic interaction way, in spite of the unavailability of AI analysis, due to either low coverage (read depth <8) or homozygosity in most or all samples for the 13 SNPs. Epigenetically, the SLE-associated SE has physical interactions with adjacent genes, including *BLK* and *FDFT1*, and the risk rs1047643-residing region. This indicates a potentially complex role of the variant rs1047643 for broad regulation by physically contacting the SE. Thus, our data provide new insights into the molecular mechanisms by merging genetic susceptibility with epigenetic impacts on gene expression for autoimmune diseases.

The *FDFT1* is a gene encoding for squalene synthase, the enzyme that catalyzes the early step in the cholesterol biosynthetic pathway (*Tozawa et al., 1999*). Previous studies have shown dyslipidemia, with elevations in total cholesterol, low-density lipoprotein, triglyceride levels in patients with lupus (*Tisseverasinghe et al., 2006*), especially in the active disease. Our multi-omics data indicate that the SNP rs1047643-linked FDFT1 may be aberrantly activated in B cell development in SLE patients, thereby providing an insight into the genetic implication of lipid metabolism for autoimmune diseases.

The limitations of this study include, due to the presence of six SLE GWAS tagging SNPs in SE region, we are unclear how they genetically influence the SE activity during B cell development. Second, it remains unclear how the AI pattern occurs in naive B cells with lupus. The C allele shows more open chromatin state in SLE naive B cells, this can't be explained by STAT3 allelic DNA binding at the T allele. This implies that some other factors may also contribute to this dynamic AI pattern.

Third, no functional studies on genetic manipulation at the rs1047643 prevent us draw the further conclusion about whether and how the rs1047643 impact the STAT3 binding in the present study.

In addition, it should be noted is the implementation of linear regression model in the initial step to identify the allelic difference signals between SLE and controls. Due to small sample size, there is no statistical power to analyze with more optimal statistical models, such as the logistic regression model. Meanwhile, the unavailability of other variables in regression analysis further restricts this study to remove the potential confounding factors. Together, our analysis may miss some potential AI signals and disable to evaluate the causation.

In conclusion, we identified a novel functional variant and B-cell-specific SE in association with the SLE pathogenesis, both mediated by STAT3, and influencing their gene targets. This insight into the mechanism by which manipulation of STAT3 affects the SE activity and its associated gene expression in B cells may have implications for future drug development in autoimmunity.

# Materials and methods

## Key resources table

| Reagent type (species) or resource | Designation | Source or reference | Identifiers | Additional information |
|---|---|---|---|---|
| Chemical compound, drug | ML115 | Cayman Chemical | Cayman Chemical: 15,178 | *Madoux et al., 2010* |
| Chemical compound, drug | S3I-201 | Sigma-Aldrich | Sigma-Aldrich: SML0330 | |
| Chemical compound, drug | Cucurbitacin I | Sigma-Aldrich | Sigma-Aldrich: C4493 | |
| Chemical compound, drug | Recombinant human IL-6 | Cell Guidance Systems | Cell Guidance Systems: GFH10AF | |
| Antibody | Phospho-STAT3 (Ser727) | Thermo Fisher Scientific | Thermo Fisher Scientific Cat# PA5-17876; RRID:AB_10980044 | |
| Antibody | Anti-Histone H3 (acetyl K27) | Abcam | Abcam Cat# ab4729; RRID:AB_2118291 | |
| Antibody | H3K4me1 Recombinant Polyclonal Antibody | Thermo Fisher Scientific | Thermo Fisher Scientific Cat# 710795; RRID:AB_2532764 | |
| Antibody | normal mouse IgG | Santa Cruz Biotechnology | Santa Cruz Biotechnology Cat# sc-2025; RRID:AB_737182 | |
| Antibody | normal rabbit IgG | Santa Cruz Biotechnology | Santa Cruz Biotechnology Cat# sc-2027; RRID:AB_737197 | |
| Cell line (*H. sapiens*) | GM11997 | Coriell | Coriell Cat# GM11997; RRID:CVCL_5C55 | |
| Sequence-based reagent | ChIP-qPCR primers | This paper | | See *Supplementary file 5* |
| Sequence-based reagent | RT-qPCR primers | This paper | | See *Supplementary file 5* |

*Continued on next page*

*Continued*

| Reagent type (species) or resource | Designation | Source or reference | Identifiers | Additional information |
|---|---|---|---|---|
| Sequence-based reagent | Allelic qPCR primers | This paper | | See *Supplementary file 5* |
| Software, algorithm | R | R Foundation | https://www.r-project.org | Version 4.0.2 |
| Software, algorithm | Hisat2 | *Kim et al., 2019* | | Version 2 |
| Software, algorithm | Allelic imbalance analysis and plots | This paper (*Zhang, 2021*) | | The R code used for the AI analysis can be accessed via github at https://github.com/youngorchuang/Allelic-imbalance-analysis, (copy archived at swh:1:rev:f0db42af8fed130ebbfe0b46abf992300dadddd6) |
| Software, algorithm | HiCUP | *Wingett et al., 2015* | | |
| Commercial assay or kit | Mycoplasma detection kit | Sigma-Aldrich | Sigma-Aldrich:MP0025 | |
| Commercial assay or kit | SuperScript III reverse transcriptase | Thermo Fisher Scientific | Thermo Fisher Scientific:18080044 | |
| Commercial assay or kit | Luna Universal qPCR Master Mix | New England Biolabs | New England Biolabs:M3003X | |

## Data collection

We collected a variety of functional genomics data, including ATAC-seq, RNA-seq, reduced-representation bisulfite sequencing (RRBS), Hi-C data (*Supplementary file 1*), from the Gene Expression Omnibus (GEO) and ArrayExpress databases. Meanwhile, we downloaded genotype and Epidemiological data from a SLE case-control study (accession: phs001025.v1) in Hispanic population (1393 cases and 886 controls) from the dbGaP database with approval (accessed 29 Sep 2020).

## Analysis of RNA-Seq and ATAC-Seq data

RNA-seq data were analyzed as described previously with few modifications (*Zhang et al., 2016*). In brief, raw sequencing data were mapped to the human reference genome (hg19) using Hisat2 program (*Kim et al., 2019*) with the default setting. Aligned data were processed and converted into BAM files using SAMtools program (*Li et al., 2009*). To quantify gene expression levels, read counts were calculated using the featureCounts (version 2.0.2) program, then implemented in the edgeR package to calculate the count per million (CPM) values.

We used a similar method described previously with several modifications (*Zhang et al., 2015*) to process the ATAC-seq data. In brief, raw sequencing data were mapped to the human reference genome (hg19) using Bowtie2 program (*Langmead and Salzberg, 2012*) with the default setting. Tag per million (TPM) metric, a method commonly used for read counting normalization, was used to quantitatively present the enrichment of open chromatin states across regions of interest.

## Identification of allelic imbalance sites

We used a similar approach described in our previous study to call variants and allelic analysis for both RNA-seq and ATAC-seq data (*Zhang et al., 2020a*). Briefly, the deduplicated reads in BAM format were realigned and recalibrated, and genetic variants were called in a multiple-sample joint manner implemented in the GATK toolkit (version 3.3). We next filtered out variants as follows: (*Parker et al., 2013*) mapping quality score <20, (*Whyte et al., 2013*) ≥ 3 SNPs detected within 10 bp distance, (*Vahedi et al., 2015*) variant confidence/quality by depth <2, (*Hnisz et al., 2013*) strand bias score >50, (*Decker and Kovarik, 2000*) genotype score <15 and (*Levy and Darnell, 2002*) read depth <8. Then,

we extracted SNPs annotated from dbSNP (Build 150) that were called as heterozygotes for each sample. For a reasonable comparison, those heterozygous SNPs identified at least triple in both case and control samples were retained for further analysis.

For a given heterozygous SNP, we calculated allelic ratio (AR) based on read coverage onto two alleles. For RNA-seq data, the resulting AR values were used to compare the AI difference of RNA transcripts between cases and controls. For ATAC-seq data, by testing for associations between AR of each heterozygous SNP (as response variable) and SLE disease status as categorical variable (case/control comparison, control and case are coded as 0 and 1, respectively) implemented in the regression analysis (see below), we analyzed the AI difference of chromatin accessibility between cases and controls. Then, the p-value and beta coefficient were calculated to estimate the significance of the association, and the differences between cases and controls, respectively.

## Genetic association analysis

For genotype data from a SLE case-control study in Hispanic population, all typed SNPs in chromosome eight were extracted for imputation using TOPMed Imputation Server (*Das et al., 2016*). To test SNP rs1047643 in association with SLE, we used a method described previously for univariate and haplotype analyses (*Shi et al., 2016*). In brief, the per-allele odds ratio (OR) and 95% confidence interval (CI) for the rs1047643 was estimated for SLE risk using a log-additive logistic model with covariates of five countries of the Hispanic population, sex and five principal components (PCs). We used the haplo.stats package in R for haplotype analyses with five countries of the Hispanic population, sex and five PCs as covariates. For SLE GWAS data in European population from *Bentham et al., 2015* study, we downloaded summary statistical data (Accession ID: GCST003156) from GWAS catalog (*Buniello et al., 2019*) and extracted statistical results for the SNP rs1047643.

A dataset of GWAS leading SNPs was downloaded from the GWAS Catalog (*Buniello et al., 2019*). Then we extracted SLE-associated SNPs at 8p23. For each indexed SLE-associated SNP at 8p23, we tested the linkage disequilibrium (LD) score ($r^2$) with query SNP rs1047643 from the data set of the Phase 3 of the 1,000 Genomes Project in European population using LDlink web tool (*Machiela and Chanock, 2015*).

## Super-enhancer annotation

We downloaded whole-genome chromatin state segmentation data (core 15-state model) for 127 cell types from the Roadmap project. As *Parker et al., 2013* defined, we consider contiguous genomic region marked by states 6–7 (enhancer states, annotated by chromHMM) with ≥3 kb as SE in a cell type. Then, we extracted and annotated super-enhancers on 8p23 locus.

## Analysis of eQTL data

We collected eQTL data sets from three large-scale studies, the Genotype-Tissue Expression (GTEx, v8) (*Laboratory, 2017*), the Haploreg v4.1 dataset (*Ward and Kellis, 2016*) and the study by *Westra, 2013*. By searching for the SNP rsID or the coordinate, we extracted the linked genes with query SNPs and plotted the results based on the significance and studies.

## Hi-C data analysis

For in situ Hi-C dataset (Accession ID: GSE63525), we downloaded the Hi-C binary file from Rao et al. study (*Rao et al., 2014*) and extracted the observed long-range interactions normalized with Knight-Ruiz matrix balancing (KR) method at 10 kb resolution across the 8p23.1 region (the coordinate: chr8:11260000–11740000, hg19).

For other genome-wide Hi-C (Accession ID: GSE113405) and capture Hi-C (CHi-C) datasets (Accession ID: GSE81503 and E-MTAB-6621), we used the Hi-C Pipeline (HiCUP) (*Wingett et al., 2015*) to truncate and align reads to the human reference genome. The deduplicated data were then processed using the Homer pipeline (*Heinz et al., 2010*) to call the significant chromatin interaction at 10 kb resolution with the support of ≥5 reads and p ≤ 0.001. The resulting interactions were visualized using UCSC Genome Browser or Sushi package in R environment.

## DNA methylation analysis

We downloaded the processed RRBS dataset of DNA methylation profiles on each CpG site from *Scharer et al., 2019* report, then extracted and compared CpG methylation levels on a region of interest between SLE and healthy controls.

## Cell culture

GM11997 B lymphoblastic (purchased from Coriell Institute) cells were cultured in RPMI-1640 medium, supplemented with 10% FBS (Thermo Fisher Scientific), 2 mM L-glutamine and 1% penicillin-streptomycin at 37 °C with 5% $CO_2$. These cells are mycoplasma-negative when tested with PCR-based mycoplasma detection kit. For perturbation of STAT3, B cells were plated in 12-well plates or 10 cm dishes one day prior to the experiment. Cells were then treated with S3I-201, ML115, Cucurbitacin I or IL-6. Cells were harvested, washed with PBS and analyzed for proper assays.

## Reverse transcription qPCR

Total RNA was isolated from cells using TRIzol Reagent (Invitrogen) according to the manufacturer's protocol. Oneμg of total RNA was reverse transcribed using SuperScript III reverse transcriptase and random hexamer. One-tenth of the RT reaction was used as a template for real-time PCR using Luna Universal qPCR Master Mix (New England Biolabs) on a QuantStudi six system. Relative expression was calculated with $2^{-\Delta\Delta Ct}$ using the average value of housekeeping gene *GAPDH*.

## Chromatin immunoprecipitation

ChIP was performed as described previously (*Whyte et al., 2013*). Approximately $10 \times 10^6$ suspension cells were harvested and in 10 ml PBS with 1% formaldehyde for 10 min at room temperature, followed by adding 0.125 M glycine for 5 min. Cells were washed and pelleted by centrifugation and lysed with buffer (50 mM Tris-HCl, pH 7.5, 1% IGEPAL CA-630, 1 mM EDTA, 0.1% SDS, plus 1 mM PMSF) in the presence of protease inhibitors and incubated on ice for 30 min. Cell lysate was sonicated to shear DNA to a length of 200–600 bp. The lysates were centrifuged, and supernatant transferred to new tubes. For immunoprecipitation, approximately $2 \times 10^6$ cells and 2–3 μg of antibodies or isotype matched IgG as control were used per ChIP and incubated with supernatant at 4 °C on a rotating wheel overnight. Chromatin-antibody complexes were sequentially washed with low-salt buffer, high-salt buffer, LiCl buffer, and TE buffer. Cross-links were reversed by addition of 100 μl of 1% SDS plus 100 mM $NaHCO_3$ and by heating at 65 °C overnight. Following phenol/chloroform/isoamyl alcohol extraction, immunoprecipitated DNA was precipitated with isopropyl alcohol and resuspended in nuclease-free water. For the identification of the specific regions of interest, ~ 10 ng of purified DNA was quantified to determine the percentage of each analyzed region, as well as a negative control region from *Fullwood et al., 2009*, against input DNA. The PCR primers are shown in *Supplementary file 5*.

## Additional statistical analysis

Data were presented as mean ± standard deviation (SD) of three replicates unless stated otherwise. Correlation analysis was performed using Pearson's correlation coefficient. Statistical significance was considered at two-sided P-values less than 0.05.

## Acknowledgements

We thank Dr. Le Su for insightful suggestions, technical discussions and critical reading of the manuscript. This study was supported by the HudsonAlpha institutional funds. The funders had no role in study design, data collection and analysis, decision to publish or preparation of the manuscript. Funding This study was supported by the HudsonAlpha Institute Fund. The funders had no role in study design, data collection and analysis, decision to publish, or preparation of the manuscript.

## Additional information

### Funding

| Funder | Grant reference number | Author |
|---|---|---|
| HudsonAlpha Institute for biotechnology funds | | Yanfeng Zhang<br>Devin M Absher |

The funders had no role in study design, data collection and interpretation, or the decision to submit the work for publication.

### Author contributions

Yanfeng Zhang, Conceptualization, Data curation, Formal analysis, Investigation, Methodology, Project administration, Resources, Software, Supervision, Validation, Writing – original draft, Writing – review and editing; Kenneth Day, Formal analysis, Investigation, Methodology, Writing – original draft, Writing – review and editing; Devin M Absher, Data curation, Funding acquisition, Investigation, Resources, Supervision, Writing – review and editing

### Author ORCIDs

Yanfeng Zhang (iD) http://orcid.org/0000-0002-3859-3839

### Decision letter and Author response

Decision letter https://doi.org/10.7554/eLife.72837.sa1
Author response https://doi.org/10.7554/eLife.72837.sa2

## Additional files

### Supplementary files

• Supplementary file 1. Summary of data sets used in the study. Functional genomics data sets, including ATAC-seq, RNA-seq and RRBS-seq data sets from seven SLE case-control studies (*Supplementary file 2*), and Hi-C data sets in multiple cell lines, and a SNP microarray data set from a lupus GWAS study.

• Supplementary file 2. List of data sets from seven SLE case-control studies.

• Supplementary file 3. Association results for the SNP rs1047643 with SLE risk in European population.

• Supplementary file 4. LD score ($r^2$) between SNP rs1047643 and 12 GWAS tag SNPs in European population.

• Supplementary file 5. List of primers used in this study.

• Transparent reporting form

### Data availability

All data generated or analysed during this study are included in the manuscript and supporting file; Source Data files have been provided for Figures 2-5.

The following previously published datasets were used:

| Author(s) | Year | Dataset title | Dataset URL | Database and Identifier |
|---|---|---|---|---|
| Scharer CD, Boss JM | 2019 | Accessible chromatin profiles of B cell subsets from healthy and SLE subjects | https://www.ncbi.nlm.nih.gov/geo/query/acc.cgi?acc=GSE118253 | NCBI Gene Expression Omnibus, GSE118253 |
| Scharer CD, Boss J | 2016 | Effects of biobanking on chromatin accessibility | https://www.ncbi.nlm.nih.gov/geo/query/acc.cgi?acc=GSE71338 | NCBI Gene Expression Omnibus, GSE71338 |

*Continued on next page*

*Continued*

| Author(s) | Year | Dataset title | Dataset URL | Database and Identifier |
|---|---|---|---|---|
| Mistry P, Nakabo S, O'Neil L, Goel RR, Jiang K, Gupta S, Dell'Orso S, Gutierrez-Cruz G, Sun HW, Kaplan MJ | 2019 | Transcriptomic, epigenetic and functional analyses implicate neutrophil diversity in the pathogenesis of systemic lupus erythematosus [ATAC-seq] | https://www.ncbi.nlm.nih.gov/geo/query/acc.cgi?acc=GSE139359 | NCBI Gene Expression Omnibus, GSE139359 |
| Caielli S, Veiga DF, Domic B, Murat E, Banchereau R, Xu Z, Chandra M, Athale S, Chung C, Walters L, Baisch J, Wright T, Punaro M, Ucar D, Ueno H, Zhou J, Banchereau J, Pascual V | 2018 | A novel CD4 T cell population expanded in SLE blood provides B cell help through IL10 and succinate [ATAC-seq] | https://www.ncbi.nlm.nih.gov/geo/query/acc.cgi?acc=GSE110017 | NCBI Gene Expression Omnibus, GSE110017 |
| Scharer CD, Boss JM | 2019 | Transcriptome profiles of B cell subsets from healthy and SLE subjects | https://www.ncbi.nlm.nih.gov/geo/query/acc.cgi?acc=GSE118254 | NCBI Gene Expression Omnibus, GSE118254 |
| Sanz I, Jenks S, Marigorta UM | 2018 | Gene expresison studies of lupus and healthy B cell subsets through RNA sequencing | https://www.ncbi.nlm.nih.gov/geo/query/acc.cgi?acc=GSE92387 | NCBI Gene Expression Omnibus, GSE92387 |
| Scharer CD, Boss JM | 2019 | DNA methylation profiles profiles of B cell subsets from healthy and SLE subjects | https://www.ncbi.nlm.nih.gov/geo/query/acc.cgi?acc=GSE118255 | NCBI Gene Expression Omnibus, GSE118255 |
| Rao S, Huntley M, Lieberman Aiden E | 2014 | A three-dimensional map of the human genome at kilobase resolution reveals prinicples of chromatin looping | https://www.ncbi.nlm.nih.gov/geo/query/acc.cgi?acc=GSE63525 | NCBI Gene Expression Omnibus, GSE63525 |
| Walter J, Manke T | 2018 | HepG2 Hi-C | https://www.ncbi.nlm.nih.gov/geo/query/acc.cgi?acc=GSE113405 | NCBI Gene Expression Omnibus, GSE113405 |
| Cairns J, Freire-Pritchett P, Wingett SW, Várnai C, Dimond A, Plagnol V, Zerbino D, Schoenfelder S, Javierre B, Osborne C, Fraser P, Spivakov M | 2016 | CHiCAGO: Robust Detection of DNA Looping Interactions in Capture Hi-C data | https://www.ncbi.nlm.nih.gov/geo/query/acc.cgi?acc=GSE81503 | NCBI Gene Expression Omnibus, GSE81503 |
| Wells AD, Chesi A, Manduchi E, Johnson ME, Leonard ME, Romberg ND, Grant SFA, Lu S | 2020 | Promoter capture-C of primary human T Follicular Helper (TFH) cells and naive CD4-positive helper T cells from tonsils of healthy volunteers | https://www.ebi.ac.uk/arrayexpress/experiments/E-MTAB-6621/ | ArrayExpress, E-MTAB-6621 |
| Disease-Specific (Systemic lupus erythematosus, NPU, MDS, RD) | 2017 | GWAS in an Amerindian Ancestry Population Reveals Novel Systemic Lupus Erythematosus Risk Loci and the Role of European Admixture | https://www.ncbi.nlm.nih.gov/projects/gap/cgi-bin/study.cgi?study_id=phs001025.v1.p1 | NCBI dbGAP, phs001025.v1 |

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
