## [Editor Report]

Through extensive computational data analyses and functional experiments, Zhang and colleagues reported a novel functional germline variant at the risk locus 8p23 for systemic lupus erythematosus. They provided evidence that the observed risk association in this locus may be mediated through the variant regulating a nearby susceptibility gene. This study advances the understanding of the genetic susceptibility and biology underlying systemic lupus erythematosus.

---

## [Decision Letter]

**Decision letter after peer review:**

Thank you for submitting your article "STAT3-mediated allelic imbalance of novel genetic variant rs1047643 and B cell specific super-enhancer in association with systemic lupus erythematosus" for consideration by *eLife*. Your article has been reviewed by 3 peer reviewers, one of whom is a member of our Board of Reviewing Editors, and the evaluation has been overseen by Betty Diamond as the Senior Editor. The reviewers have opted to remain anonymous.

Essential revisions:

1) The authors should explore large-scale GWAS data or provide any other statistical evidence to strengthen their discovery of the risk association signal of rs1047643 (such as summary statistics of GWAS data of European descendants).

2) To improve and clarify the statical analyses as suggested by reviewers.

*Reviewer #1 (Recommendations for the authors):*

This is a well conducted study aiming to identify novel risk variants for systemic lupus erythematosus risk and to further explore potential mechanisms through a comprehensive analysis of functional genomic analyses and functional assays. The authors have identified a putative novel risk variant, rs1047643, for the disease risk, and further revealed underlying disease mechanisms of a known susceptibility factor, STAT3, mediated Super enhancer activity and cis-regulatory effects of SNP rs1047643. Their discovered mechanisms are novel, providing new insights into the potential mechanism underlying other GWAS-identified variants located in the Super enhancer element to mediate the disease risk. This work could be very important to advance the understanding of the genetic susceptibility and biology of systemic lupus erythematosus risk. I hope my comments below could help authors improve their work in their revision.

1) The authors could try to explore large-scale GWAS data to strengthen their discovery of the risk association signal of rs1047643 (such as summary statistics of GWAS data of European descendant). The authors should present the information of Linkage Disequilibrium between the identified variant and previously reported GWAS-identified risk variants. The novel risk variant, rs1047643, may not be an independent risk signal (may be in LD with rs17807624).

2) Need to clarify some statistical analyses. For example, for formula presented in the line 121, it's unclear for the "disease" variable (referring to case/control status? how to link SNP information?). If referring to case/control status, it could be a response variable? The authors need to provide detailed description about how to select the risk variant rs1047643 for your focus (only this SNP satisfied your analysis a two-stage study presented in Figure 1?).

In the line 132, not sure why populations is needed to be adjusted here (only focusing on Hispanic population). The authors need to include sentence to describe about how to perform additional analysis and how to perform "AI of RNA transcripts" (the line 233/234). "Statistical analysis" section may not be necessary (including above analysis).

3) Some limitations to be included in the Discussion sections: some functional data may not be generated from the disease target cells (i.e. Bcells). The authors need to discuss some of their allelic imbalance findings about other GWAS-identified risk variants at the locus, 8p23.

4) Need to fix some typos and other minor issues. In the abstract, "SLE "- need to spell out; "Open chromatin" may change to "chromatin accessibility". A typo 'tying' (line 60); RRBS (line 87) and "SD" (line 202) need to be spelled out. The sentence about "The differences" is unclear (line 204)

*Reviewer #2 (Recommendations for the authors):*

Page 7, line 121. Formula looks awkward. An equivalent way to write this model may be Disease ~ \α + \β Allelic ratio +\epsilon. However, after formatting the model into this shape, it appears that a logistic regression for binary trait should be used instead.

Figure 2A: For your Allelic Imbalance analysis, how many SNPs have you tested in order to identify rs1047643? Should we conduct multiple test correction? I can't see any descriptions from line 108 to 124. If you have done this already (hence the p-value listed in the figure is the after-correction value), would you be clear? What is the meaning of "summary" in the last row?

Page 12 -13, the paragraph for association analysis indeed mentioned an "adjustment". I am a bit confused here: (1) If your analysis starts with Allelic Imbalance analysis, then the next follow-up association test doesn't need a multiple-test adjustment. (2) based on my question above, would you clarify what "adjustment" you are talking about and what is the mathematical procedure? In the event you did some rigorous multiple-test correction here, but not in the AI analysis, your flow of the discovery might be reversely presented.

Page 14, Line 276-284. The Hi-C loop looks interesting, however little details are provided here. E.g., what is the sequencing coverage evidencing this loop?

Figure 4 indeed shows the difference between SLE and controls. However, to make the discovery more convincing, would you check how significant is the difference with the background distribution? E.g., randomly select a hundred other sites and see whether there are also difference between SLE and controls? Or the background looks similar, putting the focal discovery in a significant (or at least promising) position? Similar concern may also apply to Figures 5 and 6: it may be nice to see how likely such difference will show up in randomly selected candidates.

*Reviewer #3 (Recommendations for the authors):*

Overall, the report by Jin Men et al. provided supportive evidence that by mining multi-omics profiles to develop and indicate a novel functional variant, the mechanism of STAT3 medication. The analytical and statistical framework is appropriate, although not fully described. In addition, additional experiments should be performed.

1. From the results presented (line 224), it is not clear the result of significance for other SNPs in the comparison. Because it has no comprehensive results, which looked like a little confusing. It's necessary to provide the list of potential significant SNPs.

2. The authors report that 'statistical significance' (line 244) about association of rs1047643 with SLE risk, and adjusted P should be 0.023 according to Figure 3. And next the choice of two SNPs for haplotype analysis, the description of the results is inadequate.

3. In Figure 6, the authors performed chip assay to confirm stat3 binding to SE and rs1047643 regions. In addition, they used the stat3 DNA binding inhibitor. However, the inhibitor didn't affect stat3 DNA binding efficiency too much. Please explain why. It's necessary to perform time-course and concentration-dependent experiments to confirm the binding specificity. Also, the authors should perform chip-PCR assay using non-specific primers. Furthermore, the authors should design the experiments to confirm whether rs1047643 affects stat3 binding.

4. The results of Figure 6b,c is confused.

[Editors' note: further revisions were suggested prior to acceptance, as described below.]

Thank you for resubmitting your work entitled "STAT3-mediated allelic imbalance of novel genetic variant rs1047643 and B cell specific super-enhancer in association with systemic lupus erythematosus" for further consideration by *eLife*. Your revised article has been evaluated by Betty Diamond (Senior Editor), a Reviewing Editor, and the original reviewers.

The manuscript has been improved but there are some remaining issues that need to be addressed: please provide a response to the comments of Reviewer #2.

*Reviewer #1 (Recommendations for the authors):*

The authors have adequately addressed all of my concerns/comments.

*Reviewer #2 (Recommendations for the authors):*

In the revised manuscript, most of my comments have been satisfactorily addressed, with the following exceptions:

Q1: you have not responded to my comments on why putting the disease status on the right-hand side of the model.

Q2: disagree that one doesn't have to conduct multiple test adjustment simply due to the sample size being small.

*Reviewer #3 (Recommendations for the authors):*

I am satisfied with the revised manuscript and suggest the manuscript to be accepted for publication.

[Editors' note: further revisions were suggested prior to acceptance, as described below.]

Thank you for resubmitting your work entitled "STAT3-mediated allelic imbalance of novel genetic variant rs1047643 and B cell specific super-enhancer in association with systemic lupus erythematosus" for further consideration by *eLife*. Your revised article has been evaluated by Betty Diamond (Senior Editor) and a Reviewing Editor.

The manuscript has been improved but there are some remaining issues that need to be addressed, as outlined below:

Please address reviewer's Q1: "you have not responded to my comments on why putting the disease status on the right-hand side of the model"

The editor agrees with the validity of the reviewer's comments and would suggest using a logistic regression with disease status as a dependent variable.

---

## [Author Response]

Essential revisions:1) The authors should explore large-scale GWAS data or provide any other statistical evidence to strengthen their discovery of the risk association signal of rs1047643 (such as summary statistics of GWAS data of European descendants).2) To improve and clarify the statical analyses as suggested by reviewers.

We thank two editors for giving us an opportunity to submit the revised paper. In the revised version of manuscript, we have made a significant improvement to address these two essential comments (Page 21, and Methods section), and have now met the high standard of your journal for publication. Please see below are the point-by-point response to the reviewers’ comments.

Reviewer #1 (Recommendations for the authors):This is a well conducted study aiming to identify novel risk variants for systemic lupus erythematosus risk and to further explore potential mechanisms through a comprehensive analysis of functional genomic analyses and functional assays. The authors have identified a putative novel risk variant, rs1047643, for the disease risk, and further revealed underlying disease mechanisms of a known susceptibility factor, STAT3, mediated Super enhancer activity and cis-regulatory effects of SNP rs1047643. Their discovered mechanisms are novel, providing new insights into the potential mechanism underlying other GWAS-identified variants located in the Super enhancer element to mediate the disease risk. This work could be very important to advance the understanding of the genetic susceptibility and biology of systemic lupus erythematosus risk. I hope my comments below could help authors improve their work in their revision.1) The authors could try to explore large-scale GWAS data to strengthen their discovery of the risk association signal of rs1047643 (such as summary statistics of GWAS data of European descendant). The authors should present the information of Linkage Disequilibrium between the identified variant and previously reported GWAS-identified risk variants. The novel risk variant, rs1047643, may not be an independent risk signal (may be in LD with rs17807624).

We added these results in the revised manuscript (Page 21, page 7: lines 123-130).

2) Need to clarify some statistical analyses. For example, for formula presented in the line 121, it's unclear for the "disease" variable (referring to case/control status? how to link SNP information?). If referring to case/control status, it could be a response variable? The authors need to provide detailed description about how to select the risk variant rs1047643 for your focus (only this SNP satisfied your analysis a two-stage study presented in Figure 1?).In the line 132, not sure why populations is needed to be adjusted here (only focusing on Hispanic population). The authors need to include sentence to describe about how to perform additional analysis and how to perform "AI of RNA transcripts" (the line 233/234). "Statistical analysis" section may not be necessary (including above analysis).

In the revised paper, we provide detailed description about AI analysis implemented in a linear model (Page 20-21), and re-clarify the adjustment of five countries of the Hispanic population (Page 21: lines 396-400), as well as the method section (Page 21).

3) Some limitations to be included in the Discussion sections: some functional data may not be generated from the disease target cells (i.e. Bcells). The authors need to discuss some of their allelic imbalance findings about other GWAS-identified risk variants at the locus, 8p23.

Thanks for reviewer’s suggestion. In the revised paper, we discussed the limitations (Page 16, lines 329-332) and the allelic imbalance results of 13 GWAS SNPs (Page 15, lines 307-308).

4) Need to fix some typos and other minor issues. In the abstract, "SLE "- need to spell out; "Open chromatin" may change to "chromatin accessibility". A typo 'tying' (line 60); RRBS (line 87) and "SD" (line 202) need to be spelled out. The sentence about "The differences" is unclear (line 204)

Thank reviewer for pointing out these issues. We corrected and defined these terms in the revised paper, except for ‘tying’ that is correct, not a typo.

Reviewer #2 (Recommendations for the authors):Page 7, line 121. Formula looks awkward. An equivalent way to write this model may be Disease ~ \α + \β Allelic ratio +\epsilon. However, after formatting the model into this shape, it appears that a logistic regression for binary trait should be used instead.

In the revised paper, we rephrased the description about AI analysis implemented in a linear model (Page 20, lines 379-385)

Figure 2A: For your Allelic Imbalance analysis, how many SNPs have you tested in order to identify rs1047643? Should we conduct multiple test correction? I can't see any descriptions from line 108 to 124. If you have done this already (hence the p-value listed in the figure is the after-correction value), would you be clear? What is the meaning of "summary" in the last row?

In the revised paper, we provided more detailed description and re-clarified the statistical analyses (Page 6, lines 93-102).

Page 12 -13, the paragraph for association analysis indeed mentioned an "adjustment". I am a bit confused here: (1) If your analysis starts with Allelic Imbalance analysis, then the next follow-up association test doesn't need a multiple-test adjustment. (2) based on my question above, would you clarify what "adjustment" you are talking about and what is the mathematical procedure? In the event you did some rigorous multiple-test correction here, but not in the AI analysis, your flow of the discovery might be reversely presented.

Thanks for reviewer’s helpful comments. In the revised paper, we provided detailed description and re-clarified the statistical analyses (Page 34, lines 682-684, and Figure 3A-B).

Page 14, Line 276-284. The Hi-C loop looks interesting, however little details are provided here. E.g., what is the sequencing coverage evidencing this loop?

In the revised paper, we provided more details about read supports and statistical significance for DNA loops in the Methods section (Page 23, lines 443-444).

Figure 4 indeed shows the difference between SLE and controls. However, to make the discovery more convincing, would you check how significant is the difference with the background distribution? E.g., randomly select a hundred other sites and see whether there are also difference between SLE and controls? Or the background looks similar, putting the focal discovery in a significant (or at least promising) position? Similar concern may also apply to Figures 5 and 6: it may be nice to see how likely such difference will show up in randomly selected candidates.

In the revised paper, we presented these results (Page 9-10, lines 178-180; Page 11, lines 208-212, and Figure 4—figure supplement 1and Figure 5—figure supplement 1).

Reviewer #3 (Recommendations for the authors):Overall, the report by Jin Men et al. provided supportive evidence that by mining multi-omics profiles to develop and indicate a novel functional variant, the mechanism of STAT3 medication. The analytical and statistical framework is appropriate, although not fully described. In addition, additional experiments should be performed.1. From the results presented (line 224), it is not clear the result of significance for other SNPs in the comparison. Because it has no comprehensive results, which looked like a little confusing. It's necessary to provide the list of potential significant SNPs.

Because of small sample size, we just found three potential significant SNPs. In the revised paper, we reported these SNPs (Page 6, lines 93-102, and Figure 2—figure supplement 1).

2. The authors report that 'statistical significance' (line 244) about association of rs1047643 with SLE risk, and adjusted P should be 0.023 according to Figure 3. And next the choice of two SNPs for haplotype analysis, the description of the results is inadequate.

In the revised paper, we provided detailed description and re-clarified the statistical analyses (Page 7, lines 120-130).

3. In Figure 6, the authors performed chip assay to confirm stat3 binding to SE and rs1047643 regions. In addition, they used the stat3 DNA binding inhibitor. However, the inhibitor didn't affect stat3 DNA binding efficiency too much. Please explain why. It's necessary to perform time-course and concentration-dependent experiments to confirm the binding specificity. Also, the authors should perform chip-PCR assay using non-specific primers. Furthermore, the authors should design the experiments to confirm whether rs1047643 affects stat3 binding.

Thanks for reviewer’s insightful comments. Please see below are detailed description.

For the question about “*the authors should perform chip-PCR assay using non-specific primers”*, we designed a pair of primers as negative control which has been previously used elsewhere as ChIP-qPCR quality control (Page 25, lines 474-476). We presented these new ChIP-qPCR results in the revised manuscript (Page 12, line 228, and Figure 6—figure supplement 1).

For the question about “*However, the inhibitor didn't affect stat3 DNA binding efficiency too much*”, we did more experiment by increasing S3I concentration to 100 μM. The results don’t change dramatically (Figure 6C). One reason may be that the rs1047643 region is co-bound by multiple STAT family members. Because in our latest experiment where we also did STAT1 ChIP assay under S3I treatment in GM11997 cells, to our surprise is that we observed an increase of STAT1 binding on rs1047643-T allele under the S3I treatment. That is to say, as classic JAK-STAT signaling pathway shows, there may exist three different sorts of dimers (two homo-dimers, STAT1-STAT1 and STAT3-STAT3, one hetero-dimer, STAT1-STAT3) that co-regulate this locus in a complicated manner. Therefore, inhibition of STAT3 may result in a mild effect on the rs1047643 allelic binding.

For the remaining questions, we would like to logically depict our study design on this project. At the beginning, our strategy was focused on loss-of-function of STAT3 by using siRNA and shRNA methods. However, due to the extremely low transfection efficiency (<5%) in such tough suspension cells, the experiments failed. Meanwhile, we attempted to test the impact of the rs1047643 on the STAT3 binding by transfection-associated assays, but were unsuccessful, presumably due to the difficulty of transfection in lymphoblastic cells. Eventually, we changed our strategies and switched to use STAT3 drug inhibitors and activators to determine the allelic binding performance.

In addition, we have expanded this project and acquired several novel interesting findings during the last nine months. In the revised manuscript, we present the relevant results to confirm the results by controlling upstream factors in the JAK-STAT3 signaling pathway (Page 12-13, lines 244-249, and Figure 6—figure supplement 3).

4. The results of Figure 6b,c is confused.

Thanks for reviewer’s helpful comment. To make the results more readable and understandable, we re-organized the Figure 6 and split into two Figures (Figure 6 and 7) in the revised manuscript.

[Editors' note: further revisions were suggested prior to acceptance, as described below.]

The manuscript has been improved but there are some remaining issues that need to be addressed: please provide a response to the comments of Reviewer #2.Reviewer #2 (Recommendations for the authors):In the revised manuscript, most of my comments have been satisfactorily addressed, with the following exceptions:Q1: you have not responded to my comments on why putting the disease status on the right-hand side of the model.

To test for associations between allelic ratio and SLE disease state (case/control comparison) for each heterozygous SNP, we regarded the case/control disease status as categorical variable (control and case are coded as 0 and 1, respectively) and put it on the right-hand side implemented in regression model, the analysis has been used in other studies (1, 2). In the revised manuscript, we re-phrased the regression analysis (Page 20, lines 382-389).

Q2: disagree that one doesn't have to conduct multiple test adjustment simply due to the sample size being small.

Thanks for reviewer’s suggestion. In the revised manuscript, we re-phrased this sentence (Page 6, line 95).

1. Adoue V, Schiavi A, Light N, Almlof JC, Lundmark P, Ge B, et al. Allelic expression mapping across cellular lineages to establish impact of non-coding SNPs. Mol Syst Biol. 2014;10:754.

2. Absher DM, Li X, Waite LL, Gibson A, Roberts K, Edberg J, et al. Genome-wide DNA methylation analysis of systemic lupus erythematosus reveals persistent hypomethylation of interferon genes and compositional changes to CD4^+^ T-cell populations. PLoS Genet. 2013;9(8):e1003678.

[Editors' note: further revisions were suggested prior to acceptance, as described below.]

Please address reviewer's Q1: "you have not responded to my comments on why putting the disease status on the right-hand side of the model"

We re-analyzed the data based on the logistic regression with disease status as a dependent variable. Due to small sample size, particularly just six samples carrying heterozygous SNP rs1047643 in one dataset (GSE71338), there is no statistical power to identify the significance when analyzing with the logistic regression model. In the revised manuscript, we discuss this limitation (Page 17, lines 334-340).